# Cost-Effectiveness Study of One-Stage Treatment of Chronic Osteomyelitis with Bioactive Glass S53P4

**DOI:** 10.3390/ma12193209

**Published:** 2019-09-30

**Authors:** Jan Geurts, Tom van Vugt, Eline Thijssen, Jacobus J. Arts

**Affiliations:** Department of Orthopedic Surgery, CAPHRI School of Public Health and Primary Care, Maastricht University, 6229 ER Maastricht, The Netherlands; t.vanvugt@gmail.com (T.v.V.); egj.thijssen@student.maastrichtuniversity.nl (E.T.); j.arts@mumc.nl (J.J.A.)

**Keywords:** bioactive glass, osteomyelitis, cost-effectivity

## Abstract

This study was set up to evaluate the costs of a one-stage treatment of chronic osteomyelitis using bioactive glass S53P4 versus a two-stage treatment using gentamicin-loaded PMMA beads. Furthermore, a cost-effectiveness analysis was performed from a hospital’s perspective together with the evaluation of clinical outcome. A treatment group (*n* = 25) receiving one-stage surgery with bioactive glass was retrospectively compared with a two-stage control group (*n* = 25). An assessment was made of all costs included from first outpatient visit until one year after treatment. Bootstrap simulation and sensitivity analyses were performed. The primary endpoint was cost-effectiveness with clinical outcome as the secondary endpoint. The base case analyses shows dominance of the one-stage treatment with bioactive glass S53P4 due to lower costs and a better clinical outcome. Sensitivity analyses confirm these findings. This study is the first in its kind to show one-stage treatment of chronic osteomyelitis with bioactive glass S53P4 to be cost-effective.

## 1. Introduction

Chronic osteomyelitis is still a challenging problem for surgeons and patients [1,2,3]. The treatment is time consuming, often involves multiple surgeries and complete eradication is never certain. Historically, treatment has consisted of a two-stage procedure with debridement of infected bone and soft tissues in the first operation. Intermittently, antibiotics were administered locally as well as parenterally. After a certain period, the local antibiotic spacer could be removed and a reconstruction performed of the bony defect [4,5].

Recently, new biomaterials have been introduced that can be used to fill bone defects in the presence of infection. Bioactive glass is such a novel biomaterial. S53P4 Bioactive glass (S53P4 Bonalive^®^, Bonalive, Turku, Finland) is a synthetic bone graft substitute material consisting of (wt%/mol%) 23.0/22.7 Na₂O, 20.0/21.8 CaO, 4.0/1.7 P₂O₅, 53.0/53.9 SiO₂ [6]. When exposed to a physiological environment, physicochemical reactions create a bone-like hydroxyapatite layer on the bioactive glass surface [7,8]. This surface makes the glass osteoconductive, with very strong bonds to bone being formed in vivo [7,8,9,10,11]. Additionally, the physicochemical reactions lead to a series of interactions that stimulate new bone formation [12,13,14,15,16,17,18,19,20,21,22]. Because of the aforementioned advantageous properties, bioactive glass is currently used for various clinical indications. Bioactive glass has also drawn attention for research on its unique antibacterial effects. In vitro studies showed that S53P4 bioactive glass has an antibacterial effect on a large panel of clinically important pathogens [23,24,25,26,27,28]. These findings not only mean that a S53P4 bioactive glass graft layer is unlikely to become infected, they also indicate that the material might be an adjunct to treat bacterial infections in vivo. Recent clinical studies validate the working mechanism and indeed show increasing evidence that S53P4 bioactive glass granules are effective in one-step surgical treatment of chronic osteomyelitis [29,30,31,32]. The structuring of the atoms of bioactive glass are very important for its bioactive mechanisms. The silica network is composed of bridged oxygen (BO) and non-bridged oxygen (NBO) atoms. The non-bridged oxygens contain sodium and calcium ions, and phosphate is incorporated in the structure. When exposed to an aqueous environment, a suggested mechanism will lead to the bioactive reaction. After implantation, water will enter the porous structure of the glass where it exchanges hydrogen for sodium at the NBO sites that leads to increased pH. The high local pH levels start breaking down the network due to hydrolysis of the Si-O bounds. This makes ions (Na, Ca, P) enter the surrounding fluid of the glass and elevates the osmotic pressure, which in combination with the high pH levels, has been postulated to have an anti-bacterial effect, since bacteria cannot withstand these conditions [23,24,25,26,27]. Hydrolysis of the network leads to release of silanols which condensate (when leaving the high local pH environment) on the surface of the glass to form a very porous network that recruits water and forms a solgel. This solgel is penetrable for ion exchange (surface reaction is ongoing) and the gel functions as a nucleation ground for CaP (released from the glass or present in the environment) to precipitate. The formed CaP layer thereafter crystalizes to form hydroxyapatite (HA), the main mineral component of bone. This layer uses bone ingrowth on the surface of the glass [18,20,22].

Due to a rise of healthcare costs at one hand and new medical developments on the other hand there is a constant tension between the treatment possibilities and burden on the local health care providers. Economic evaluations or in other terms cost-effectiveness studies are important to support evidence-based decision-making. There is growing awareness that cost-effectiveness is becoming increasingly important in healthcare, and therefore new treatment methods that reduce cost and improve outcome are of particular interest to (local) healthcare providers. This is also reflected in the increased number of economic evaluation studies in various medical fields [33,34].

With the development of newer biomaterials that can also be used in an infected environment, the approach to treatment of chronic osteomyelitis is also changing to a one-stage procedure. This might be beneficial from both clinical and cost-effectiveness point of view but has yet to be proven.

In this study, we evaluate the total cost and cost-effectiveness of treatment of chronic osteomyelitis with the use of bioactive glass S53P4 in a one-stage setting, compared to a group that were treated in a two-stage fashion with intermittent application of antibiotic loaded PMMA beads.

## 2. Materials and Methods

### 2.1. Guidelines

The current study and economic evaluation were reported according to the Consolidated Health Economic Evaluation Reporting Guidelines (CHEERS) [35].

### 2.2. Treatment Procedures

Historically, patients with chronic osteomyelitis of the long bones were treated in our center with a two-stage protocol. During the first surgery, bone and soft tissues were debrided extensively with removal of all necrotic tissues, sequestrae and excision of existing fistulae and the wound bed thoroughly lavaged. To obliterate the dead space and as a way of local antibiotic delivery, the bone defect was then filled with (prefabricated) gentamicin-loaded PMMA beads (polymethylmethacrylate) in all patients [36]. During the second surgery, two weeks later, the PMMA beads were removed and the bone defect filled with autologous bone graft or allograft. Adjuvant antibiotic therapy was administered, adapted to the results of the microbiology. Patients were admitted in hospital in between stages.

Since 2011, a new biomaterial was introduced at our center to treat chronic osteomyelitis, namely S53P4 bioactive glass (Bonalive^®^, Bonalive, Turku, Finland). These patients all underwent a one-stage treatment in which the debridement of the infection and definitive treatment of the bone defect were accomplished in the same operation. Similar to earlier protocol, patients would receive six weeks of concomitant antibiotic therapy, according to the results of culture specimens taken during surgery. Gradually, we introduced FDG PET-CT as it became available as a diagnostic tool in our hospital to evaluate the extent of the infection and the borders of the necessary debridement preoperatively and in some patients as a tool to confirm infection eradication postoperatively. The use of FDG PET-CT was validated for diagnosis of osteomyelitis in several publications [37,38,39,40,41]. Also, several patients in this group received a PICC-line (peripherally inserted central catheter) as a way of delivering antibiotics intravenously at home (OPAT: outpatient parenteral antibiotic therapy) whenever this was possible [42,43,44,45].

### 2.3. Study Design

A single-center retrospective comparative study including an economic evaluation was performed at the Maastricht University Medical Centre, a university medical centre situated in the South of the Netherlands. Patients included all had longstanding chronic osteomyelitis. One group of patients was treated with S53P4 bioactive glass in a one-stage fashion (the treatment cohort, *n* = 25 patients) between November 2011 and December 2015. These were compared to a control cohort (*n* = 25 patients) treated with a two-stage protocol. The latter group was all from a time period just before we started using bioactive glass, ranging from January 2006 to November 2011, and from our own institution.

There were no exclusion criteria for this study, other than diabetic foot, spinal osteomyelitis and peri-prosthetic joint infection. All patients were included consecutively. Local ethical committee approval was obtained for this study (METC 174084, Maastricht University 26 May 2017).

### 2.4. Estimating Resource Use and Costs

The cost-effectiveness study was performed from a hospital perspective with a time horizon from first intake until one year postoperative. Costs were identified for all patients, including the cost of clinical and outpatient procedures and the cost of inpatient hospital days. They are expressed in Euros. Consumer price indices were used to adjust all costs to the index year 2016. To calculate the cost, a three-step method was used. First, the resource use was measured; this was done by checking the medical records and the hospital billing administration. Secondly, the unit prices were identified in order to value the resource use. They were based on cost prices from the Dutch manual for cost research when available or otherwise on bottom-up cost price calculations. The latter method was applied for the e.g., cost of surgery, operating room use, microbiology, PET-CT and antibiotics. For this, the costs of graft and other material purchase prices, equipment purchase, personnel costs (hourly wages), and hospital overhead were included. In the third step, the resource use and the unit price were multiplied, in order to calculate all health care costs for every patient. Discounting of costs was not necessary, since the follow-up period of the study did not exceed one year.

### 2.5. Study Endpoints

Primary: cost-effectiveness

Secondary: clinical outcome after one year

### 2.6. Analytical methods

#### 2.6.1. Cost-Effectiveness and Costs

Base case analysis regarding clinical effectiveness, costs and cost-effectiveness was performed. Statistical analysis of eradication of infection was performed using the Fischer’s exact test. Univariate analysis was performed to compare baseline characteristics and to assess the influences of possible confounders, using the Fischer’s exact test and the Mann-Whitney-U test. Data were considered statistically significant if the *p*-value was <0.05. In the case of missing values, means were calculated from the number of patients with available values. This only affected 1 missing preoperative BMI value of a patient.

An incremental cost-effectiveness ratio (ICER) was calculated by dividing the incremental costs of both treatment options by the incremental effects of the two treatment options.

A bootstrap simulation was performed (this is a non-parametric method of cost-effectiveness analysis, as data are generally not distributed normally). This method estimates the distribution of costs and cost-effectiveness outcomes by relying on random sampling with replacement (in this study 5000 replications were performed) [46,47,48]. Bootstrap results are presented as a 95%-confidence interval (CI). The ICERs were presented in a cost-effectiveness plane (CE-planes) (Figure 1).

#### 2.6.2. Sensitivity Analyses

Results of different outcomes were tested on robustness by performing four one-way and two three-way sensitivity analyses. The one-way sensitivity analyses varied the duration of the bioactive glass surgical intervention (reducing operating time up to 40% in order to reflect the maximum effects of the learning curve using this material) and excluded outliers of total cost and impact of complications respectively. The three-way analyses varied the total duration of hospital admission.

#### 2.6.3. Clinical Outcomes

Successful clinical outcome was defined as absence of clinical signs of infection and normalization of inflammatory parameters (sedimentation rate, leucocyte count, CRP). Bone defect healing was not quantifiable within the set-up of this study and as such not part of the final analysis. Statistical analysis of eradication of infection was performed using the Fischer’s exact test. Univariate analysis was performed to compare baseline characteristics and to assess the influences of possible confounders, using the Fischer’s exact test and the Mann-Whitney-U test. Data were considered statistically significant if the *p*-value was <0.05. In the case of missing values, means were calculated from the number of patients with available values. This only affected one missing preoperative BMI value of a patient.

## 3. Results

### 3.1. Patient Characteristics

In total we included 50 patients in this evaluation (34 male/16 female) with a mean age of 52.2years. On average, patients had 1.9 previous surgeries (SD = 3.3) before enrolment in either of our treatment protocols. Mean time from onset of symptoms to the first surgical procedure was 6.1 months (SD = 7.9). See Table 1 for baseline characteristics of the total study population. Groups were not significantly different (Table 2). All patients had a minimum of 12 months follow-up. Pathogenesis of the chronic osteomyelitis was post-traumatic in 34 cases (68%), implant-related in 4 (8%), haematogenous in 8 (16%) and unknown in 4 (8%). Location was mainly in the long bones: tibia in 22, femur in 21, humerus in 3 and olecranon in 1. Other locations included calcaneus in 2 and public bone in 1. Microbiology showed S. aureus in 17 cases, Pseudomonas in 3, Coagulase-negative S. in 2, Enterococcus in 1, mixed flora in 4 and 14 cases were culture negative. According to Cierny–Mader classification 7 cases were type I, 6 type II, 12 type III and 17 type IV. 19 patients were type A hosts, 21 type B and 2 type C (Table 2.2) [49]. Nature of the bone void filler in the control group was autograft (iliac crest) in 20 cases, allograft in 5.

### 3.2. Cost Analyses

The total health care costs were significantly lower (€-6573, in the S53P4 bioactive glass treatment cohort € 20568.31 versus the PMMA contro cohort € 27,131.69 with 95% CI: €12,572–€-1295). Hospital admission costs accounted for more than half of the total cost in each group. A significant decrease was found in hospitalization cost (€-7272, 95% CI: €-12,339–€-2878), surgical cost (€-960, 95% CI: €-1734–€-328) and antibiotic cost (€-527, 95% CI: €-832–€-277). In contrast, material cost increased significantly using S53P4bioactive glass (€1640, 95% CI: €1123–€2256) as well as imaging costs (€817, 95% CI: €208–€1368) (Table 3).

### 3.3. Cost-Effectiveness Analyses

The base case analyses show that the one-stage treatment with S53P4 bioactive glass is dominant, it is more effective (successfully treated patient) and at the same time results in a decrease of costs (an average of € 6573). The bootstrap analyses confirm this dominance, 93% of all ICERs are situated in the Southeast quadrant, indicating that the one-stage group is dominant over the control group. Only 6% of the bootstrapped ICERs were situated in the Southwest, 1% in the Northwest quadrant and zero percent in the Northeast quadrant (Figure 1).

The incremental cost-effectiveness ratio (ICER) was calculated to be €54,443. This implies that per successfully treated patient, a saving of €54,443 can be expected due to direct savings in cost, combined with a positive clinical outcome.

### 3.4. Sensitivity Analysis

To test the impact of outliers, patients were excluded with extreme variation in treatment (complications or additional surgeries) in the first one-way sensitivity analysis (1 in the treatment cohort, 3 in the control cohort) and patients that deviated more than 50% of the average cost in the second one-way sensitivity analysis (2 in the treatment cohort, 3 in the control cohort). Exclusion of these outliers decreased the dominance of the one-stage treatment being cost-effective, but it still remained a difference of 91% and 95% over the two-stage treatment, respectively. The same accounts for the third one-way analysis, decreasing the surgical time by 40%, resulting in a 92% change of the one-stage treatment being dominant.

In the three-way analyses, hospital stay was varied 40% (up or down), resulting in a dominance for the treatment group of 66% (cost saving of € 1589) when increasing the stay and a dominance of 93% (cost saving of €11,477) when decreasing the stay in the same amount, both thus clearly favoring the treatment with bioactive glass (Table 4).

### 3.5. Clinical Effectiveness

Eradication of infection was 92% in the S53P4 bioactive glass treatment group (23 out of 25 patients) versus 80% in the PMMA control group (20 out of 25 patients). This result was not statistically significant (*p* = 0.209; Fisher’s Exact test). Clinical outcomes showed a significantly shorter hospital stay in the S53P4 bioactive glass group (18.3 versus 32.7 days, *p* < 0.001; Mann-Whitney U). Mean number of surgical interventions decreased from 2.3 to 1.3 (*p* < 0.001; Mann-Whitney U). No significant differences were found in wound healing problems, prolonged leakage or complications. There was one major complication in the S53P4bioactive glass group (femoral fracture, requiring additional surgery) and two in the control group (1 death, 1 amputation).

Duration of stay and surgical cost were significantly lower in the S53P4 bioactive glass cohort (*p* < 0.001), whereas cost of implant material was significantly higher (*p* < 0.001).

## 4. Discussion

Chronic osteomyelitis is a problem with far reaching consequences for the patient and the health system, as morbidity and complications can be numerous and treatment intensive, both from a duration standpoint as well as a cost standpoint. New biomaterials, such as S53P4 bioactive glass, have recently been introduced and proven to be clinical effective. The antibacterial working mechanism relies on a local increase of pH and osmotic pressure around the glass granules, making the environment hostile for bacterial adhesion and proliferation [27]. S53P4 also acts as a bone substitute due to its osteoconductivity and ability to induce proliferation and differentiation of stem cells [14,50]. Very few studies to date evaluate the costs, associated with the use of state-of-the-art biomaterials like S53P4 bioactive glass, or only looked at certain elements of it.

To our knowledge, this study is the first to describe and compare treatment modalities for patients suffering from chronic osteomyelitis from a cost and cost-effectiveness point of view. Usage of S53P4 bioactive glass was described in several clinical papers with good results [32,51]. Only one other study compared clinical effectiveness of S53P4 to other biodegradable antimicrobial biomaterials, but without any cost-effectiveness evaluation [52]. Bernard et al. evaluated the cost of outpatient parenteral antimicrobial therapy (OPAT) and concluded that this treatment resulted in a substantial cost saving of $1,873,885 in a group of 39 patients, compared to in-hospital antibiotic treatment ($120 per day versus $710) [42]. Total cost of the new treatment protocol with S53P4 bioactive glass in this study was significantly reduced due to a decrease in hospital stay, fewer surgeries and lower antibiotic cost. Results of the cost-effectiveness analysis were strongly in favor of this treatment protocol, which was confirmed by the robustness of the bootstrap analysis data.

We believe our results can be achieved in other hospitals in the western world with expertise in treatment of chronic osteomyelitis. Important is the adherence to a strict protocol in order to have a treatment regime that is more or less equal for all patients involved. The one-stage concept of treating chronic osteomyelitis is, however, a rather new concept.

Limitations of this study include the retrospective analysis; however, with one-stage treatment using bioactive glass being a very successful procedure, it would not be ethical to have a control group that is treated in a suboptimal manner. Another option would be to compare our series of bioactive glass patients with a group treated in the two-stage fashion in another hospital, but no such hospital with a significant number of treated patients during the same period could be identified in the Netherlands. Therefore, we decided to take our own series of patients treated in the two-stage fashion just before introduction of the S53P4 bioactive glass as the control group. This also made the cost comparison more adequate and reliable.

Secondly, cost-effectiveness is ideally not solely based on outcomes like eradication of pathology, but also on QALYs (quality-adjusted life years). Unfortunately, quality of life was not measured as part of this study and no information on this topic with regards to chronic osteomyelitis is available in the literature.

Thirdly, the cost-effectiveness study is only performed from a hospital perspective, while it is recommended to perform this from a societal perspective (taking into account all costs irrespective who’s paying for it).

Finally, the treatment protocol in our institution was not altered significantly between the two groups, but was more strictly adhered to in the S53P4 bioactive glass group. In combination with newer treatment options, such as home-administered antibiotics, this led to a tendency of shorter hospitalization times. On the other hand, newer, more expensive imaging modalities were introduced (PET-CT) but did not result in an overall increase of cost.

## Figures and Tables

**Figure 1 materials-12-03209-f001:**
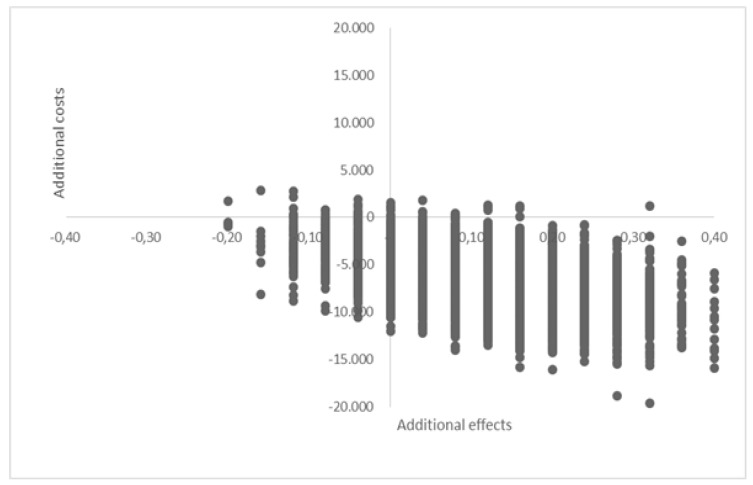
Cost-effectiveness plane of bootstrap results (*n* = 5000).

**Table 1 materials-12-03209-t001:** Baseline characteristics of the total study population.

	Total	Treatment Cohort (S53P4 Bioactive Glass)	Control Cohort (PMMA)	*p*-Value
Patients	50	25	25	
Gender (M/F)	34/16	16/9	18/7	0.595
Age (years) [range]	52.2 (SD = 17.3)	53.5 (SD = 19.8)	50.9 (SD = 14.8)	0.554
BMI (kg/m^2^) [range]	26.2 (SD = 5.4)	26.5 (SD = 6.7)	25.9 (SD =3.8)	0.687
Smoking (Y/N)	15/35	5/20	10/15	0.128
Previous surgeries	1.9 (SD = 3.3)	1.0 (SD = 2.6)	2.7 (SD = 3.7)	0.061
Time to surgery (months) [range]	6.1 (SD = 7.9)	4.0 (SD = 4.4)	8.2 (SD = 9.9)	0.057

**Table 2 materials-12-03209-t002:** Cierny–Mader classification of included patients.

	Treatment Group				Control Group				Total
	Host A	Host B	Host C	Subgroup total	Host A	Host B	Host C	Subgroup total	
Type I – Medullary	4	1	0	5	2	0	0	2	7
Type II – Superficial	0	1	0	1	3	2	0	5	6
Type III – Localized	1	4	0	5	4	3	0	7	12
Type IV – Segmental	1	4	1	6	4	6	1	11	17
Total	6	10	1	17	13	11	1	25	42

**Table 3 materials-12-03209-t003:** Average resource use and costs per category at 12-month follow-up (Bootstrap results).

Category Units	Unit Costs in Euros (€)	Treatment (S53P4) Cohort (*n* = 17)		Control (PMMA) Cohort (*n* = 25)		Average Cost Differences	
		Average Number	Mean Costs in €	Average Number	Mean COSTS in €	Cost differences	95% CI
**Hospital costs**							
Hospitalization							
Standard care unit (day)	642	18.8	12,146.95	29.0	18,533.85	−6497.04	(−10,375; 2259)
Intensive care unit (day)	2015	0.0	0	0.3	622.23	−622.23	(−1934; 0)
**Total hospital costs**		**17.9**	**11,882.14**	**29.3**	**19,154.23**	**−7272.09**	**(−12,339; −2,78)**
Surgical costs							
General surgery (minutes)	12.14	151.4	1841.88	222.3	2711.22	−869.34	(−1475; −253)
Orthopaedic surgery (min)	4.59	88.1	404.7	132.6	607.40	−203.03	(−342; −69)
Average surgical costs			2205.78		3306.72	−1100.94	(−1891; −345)
PICC-line (per placement)	569.90	0.6	344.27	0.24	162.22	182.05	(23; 342)
Fluoroscopy (per surgery)	150.69	0.2	30.43	0.4	60.29	−29.86	(−66; 6)
**Total Surgical costs**			**2574,71**		**3535.33**	**−960.62**	**(−1734; −218)**
Material costs							
S53P4 BAG (1 cc)	89	22.3	2007.20	n.a.	0	2007.20	
PMMA beads (1 bead)	6.12	n.a.	0.00	59.7	365.00	−365.00	
**Total material costs**			**2012.15**		**372.24**	**1639.91**	**(1123; 2.256)**
Imaging costs							
X-ray	49.55	4.84	240.10	9.12	450.87	−211	(−343; −93)
CT	140/145	0.56	77.40	0.88	123.44	−46	(−129; −33)
MRI	229/215	0.40	86.24	0.48	102.53	−16	(−104; 86)
FDG PET-CT	1275.82	1.28	1620.14	0.36	460.98	1159	(612; 1685)
Scintigraphy	338.50	0.12	40.76	0.32	105.94	−65	(−149; 27)
**Total imaging costs**			**1960.63**		**1142.83**	**818**	**(208; 1368)**
Blood sample analysis							
C-reactive protein	4.07	9.1	36.87	12.12	48.96	−12	(−27; 0)
Erythrocyte sedimentation	1.08	8.7	9.38	12.76	13.80	−4	(−8; −1)
Leukocyte count	1.08	6.2	6.69	12.88	13.95	−7	(−11; −4)
Sodium	1.79	9.08	16.26	8.40	15.03	1	(−6; 8)
Potassium	1.79	8.96	16.08	8.48	15.23	0.9	(−7; 8)
Creatinine	1.78	9.12	16.29	8.88	15.84	0.5	(−7; 8)
Urea	1.63	9.08	14.78	8.60	14.09	0.7	(−6; 8)
Alanine aminotransferase	2.10	2.24	4.64	3.80	7.96	−3	(−6; 0)
Aspartate aminotransferase	1.95	2.20	4.31	3.76	7.32	−3	(−5; 0)
Gammaglutyltransaminase	1.96	2.24	4.38	3.68	7.23	−3	(−6; 0)
Haemoglobin/haematocrit	1,74	10.60	18.56	12.16	21.25	−3	(−11; 6)
**Total blood analysis**			**148.40**		**179.26**	**−31**	**(−96; 28)**
Microbiological cultures							
Tissue cultures	67.38	3.7	246.25	2.4	159.25	87	(10; 160)
Fluid cultures	55.48	1.8	102.75	4.0	220.97	−118	(−20; −21)
**Total microbiology**			**346.17**		**381.67**	**−36**	**(−139; 64)**
Antibiotics							
Intravenous (day)	25.40	15.2	332.09	32.2	856.09	−524	(−848; −237)
Oral (day)	1.01	49.7	43.27	55.6	60.93	−18	(−41; 5)
**Total antibiotics**			**346.46**		**873.48**	**−527**	**(−832; −277)**
Outpatients’ Clinic costs							
**Visits**	**163**	**8.1**	**1327.06**	**8.8**	**1445.51**	**−118**	**(−424; 202)**
**Total costs**			**20,568.31 (SD 1599.37)**		**27,141.69 (SD 2410.12)**	**−6573**	**(−12,572; −1295)**

**Table 4 materials-12-03209-t004:** Results of different Sensitivity Analyses (effects applied only on the treatment group).

Cost-Effectiveness per Eradication	∆ Costs (€)	∆ Effects	ICER	Distribution of Cost-Effectiveness Plane (quadrant, %)			
				NE	NW	SE	SW
**Base case**	**−6533**	**0.12**	−54,443	0	1	93	6
Exclusion of outliers							
Treatment-based	−5926	0.08	−69,730	0	0	91	8
Cost-based	−7217	0.12	−59,508	0	0	95	5
Surgical time variance							
40% shorter	−7416	0.12	−61,799	0	0	92	8
40% longer	−5651	0.12	−47,087	1	1	92	6
Hospital stay variance							
40% shorter	−11477	0.12	−95,642	0	0	93	7
40% longer	−1589	0.12	−13,244	22	10	66	2

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
