# Peer review of "Cost-Effectiveness Study of One-Stage Treatment of Chronic Osteomyelitis with Bioactive Glass S53P4"

_materials, 2019, doi:10.3390/ma12193209_

Round 1

Reviewer 1 Report

I congratulate the authors to this clinically very important study.  Chronic osteomyelitis is a severe problem for the patient and the treatment is time and cost consuming. Improvement of the treatment possibilities is strongly needed. The cost effectiveness analysis shows in all calculation modalities a clear advantage of the one step procedure over the two stage procedure. Even if patient’s satisfaction and QALY´s are not analyzed in the study, one could speculate that the shorter hospitalization and only one surgery will have a positive impact on both parameters.

Comments:

1. Study population: There is a clear trend to more severe cases in the control group: 66% of the patients were smoker in the control group compared to 25% in the treatment group. Also more previous surgeries were done in the control group and the time until surgery as doubled. It might be that no significant differences were seen due to the rather small groups, but these points need to be discussed.

2. The outcome parameter should be defined more precise.  Was it only eradication of infection or also successful healing? A fully regenerated bone defect is important for a positive outcome and this should be included in the outcome analysis.  

3. line 126: What is meant with …surgical interventions decreased from 2.3 to 1.3? Was further operation needed in both cohorts? If yes, please provide details.

4. Was no ethic approval necessary for this study?

5. line 50: check references.

6. line 75, please abbreviate Staphylococcus with “S.” not “St.”. (2 times)

7. Conflict of interest: is none of the authors member of the clinical advisory board of BonAlive?

Author Response

Response to reviewer 1 comments

Study population: There is a clear trend to more severe cases in the control group: 66% of the patients were smoker in the control group compared to 25% in the treatment group. Also more previous surgeries were done in the control group and the time until surgery as doubled. It might be that no significant differences were seen due to the rather small groups, but these points need to be discussed.

        Indeed, groups were too small to show significant differences. Initially, the plan was to have two         randomized groups, but our ethical comittee did not allow us to subject the "controls" to a                 two-stage procedure anymore, since the use of bioactive glass in a one-stage setting has been         shown to be effective and safe. Next option would have been to compare our treatment group to         a group of patients, treated in the classical two-stage setting, but from another hospital in the             Netherlands treating osteomyelitis. No such center was identified. So, finally, we had no other         option but to compare the treatment group to our own set of patients treated two-stage, just             before the onset of ther introduction of bioactive glass in our department. These patients have a         slightly different demography and were also treated in other institutions several times before             being referred. Nowadays, patients are promptly referred and as a consequence have had less         previous surgeries.

    2. The outcome parameter should be defined more precise.  Was it only eradication of infection or         also successful healing? A fully regenerated bone defect is important for a positive outcome and         this should be included in the outcome analysis.

        Succesfull clinical outcome was defined as absence of clinical signs of infection and         normalization of inflammatory parameters (sedimentation rate, leucocyte count, CRP).         Bone defect healing was not quantifiable within the set-up of this study and as such             not part of the final analysis.

    3. line 126: What is meant with …surgical interventions decreased from 2.3 to 1.3? Was further             operation needed in both cohorts? If yes, please provide details.

        adressed

    4. Was no ethic approval necessary for this study?

        Study was indeed approved by our local etical committee (METC 174084). Adjusted in                     manuscript

    5. line 50: check references.

        adressed

    6. line 75, please abbreviate Staphylococcus with “S.” not “St.”. (2 times)

        corrected

    7. Conflict of interest: is none of the authors member of the clinical advisory board of BonAlive?

        Geurts and Arts are/became members of the CAB of BonAlive since 2015, so after the time of         the last patient included in this study was treated. More than happy to mention it, though.

Reviewer 2 Report

I find the article opportune and well written, with conclusions well supported by the study. 

My only and main concern, however, lays on the characteristics of the Bioglass and control (PMMA) group. Three of the baseline characteristics of the study population shown in Table 2.1, seem overrepresented in the control group as compared to Bioglass group. These are: Smoking, Previous surgeries and Time to surgery. Considering all three characteristics play against a positive outcome, the better results found in Bioglass group might be related to worst starting point conditions of the control group. Can the authors clarify this?

Author Response

Response to reviewer 2 comments

My only and main concern, however, lays on the characteristics of the Bioglass and control (PMMA) group. Three of the baseline characteristics of the study population shown in Table 2.1, seem overrepresented in the control group as compared to Bioglass group. These are: Smoking, Previous surgeries and Time to surgery. Considering all three characteristics play against a positive outcome, the better results found in Bioglass group might be related to worst starting point conditions of the control group. Can the authors clarify this?

        Initially, the plan was to have two randomized groups, but our ethical comittee did not allow             us to subject the "controls" to a two-stage procedure anymore, since the use of bioactive glass         in a one-stage setting has been shown to be effective and safe. Next option would have been to         compare our treatment group to a group of patients, treated in the classical two-stage setting,         but from another hospital in the Netherlands treating osteomyelitis. No such center was                     identified. So, finally, we had no other option but to compare the treatment group to our own set         of patients treated two-stage, just before the onset of ther introduction of bioactive glass in our         department. These patients have a slightly different demography and were also treated in other         institutions several times before being referred. Nowadays, patients are promptly referred and as         a consequence have had less previous surgeries. Also, we have a far more protocollised                 treatment approach nowadays, compared to when the initial (control) patients were treated.             However, as seen in the results, groups were probably still to smaal to show a significant                 improvement in outcome. Primary aim of this paper was to show cost effectiveness.